# Association between First-Generation Antihistamine Use in Children and Cardiac Arrhythmia and Ischemic Heart Disease: A Case-Crossover Study

**DOI:** 10.3390/ph16081073

**Published:** 2023-07-28

**Authors:** Ju Hee Kim, Hye Ryeong Cha, Eun Kyo Ha, Ji Hee Kwak, Hakjun Kim, Jeewon Shin, Hye Mi Jee, Man Yong Han

**Affiliations:** 1Department of Pediatrics, Kyung Hee University Medical Center, Seoul 02447, Republic of Korea; 2004052@gmail.com; 2Department of Computer Science and Engineering, Sungkyunkwan University, Suwon 16419, Republic of Korea; ryoung0156@gmail.com; 3Department of Pediatrics, Kangnam Sacred Heart Hospital, Seoul 07441, Republic of Korea; dmsry1@gmail.com; 4Department of Pediatrics, Kangbuk Samsung Hospital, Sungkyunkwan University School of Medicine, Seoul 03181, Republic of Korea; hihikwak@gmail.com; 5Department of Obstetrics and Gynecology, Hwacheon County Health and Medical Center, Hwacheon 24119, Republic of Korea; noeun001@naver.com; 6Department of Pediatrics, CHA Bundang Medical Center, CHA University School of Medicine, Seongnam 13496, Republic of Korea; jshin8@gmail.com

**Keywords:** antihistamine, cardiovascular event, arrhythmia, children, drug adverse effect

## Abstract

Cardiotoxicity from first-generation H1-antihistamines has been debated since the 1990s. However, large-scale studies on this topic in a general pediatric population are lacking. This study aimed to assess the association between first-generation H1-antihistamine use and cardiovascular events in a nationwide pediatric population. In this case-crossover study, the main cohort included children with cardiovascular events from the National Health Insurance Service database (2008–2012 births in Korea) until 2018. The second cohort excluded children with specific birth histories or underlying cardiovascular diseases from the main cohort. Cardiovascular events of interest included cardiac arrhythmia and ischemic heart disease. Odds ratios (ORs) of cardiovascular events were estimated using conditional logistic regression models, comparing first-generation H1-antihistamine use during 0–15 days before cardiovascular events (hazard period) with use during 45–60 and 75–90 days before events (control periods). Among the participants, 1194 (59.9%) were aged 24 months to 6 years, and 1010 (50.7%) were male. Cardiovascular event risk was increased among users of first-generation H1-antihistamines (adjusted OR [aOR], 1.201; 95% confidence interval, 1.13–1.27). Significant odds of cardiovascular events persisted within 10 and 5 days (aOR, 1.25 and 1.25). In the second cohort, the association was comparable with that in the main cohort. Our findings indicate that cardiovascular event risk is increased in children who are administered first-generation H1-antihistamines.

## 1. Introduction

H1-antihistamines are the most commonly administered drugs for children [1]. H1-antihistamines can relieve histamine-mediated symptoms and consequently are administered to patients with allergic disorders, such as food allergies, allergic rhinitis, chronic spontaneous urticaria, atopic eczema, and anaphylaxis [2,3]. Additionally, although it has been reported that H1-antihistamines have no clear beneficial effect on symptoms, such as nasal congestion, rhinorrhea, and sneezing caused by the common cold, they have been widely used in children to relieve these symptoms [4]. 

First-generation H1-antihistamines are widely used in pediatric patients, owing to their extensive history of long-term use [5]. However, few studies have reported their pharmacological properties and adverse effects in infants and children [6]. A significant concern associated with first-generation H1-antihistamine use in children is their potential cardiovascular side effects [7]. In the 2010s, the European Medicines Agency [8] and Health Canada [5] warned that first-generation H1-antihistamines, especially hydroxyzine, increased the risk of QT prolongation and torsade de pointes (TdP). Additionally, several cases of cardiovascular toxicity have been reported following H1-antihistamine overdose in children. 

Although cardiovascular events, including arrhythmia and myocardial infarction, are rare in young children, these conditions can have serious consequences. Therefore, it is important to investigate the potential cardiovascular risks associated with first-generation H1-antihistamines. However, there is a paucity of studies confirming whether commonly prescribed first-generation H1-antihistamines increase the risk of cardiovascular events in young children.

To address this knowledge gap, we conducted a nationwide observational cohort study with a case-crossover design using administrative data obtained from the general pediatric population. This study aimed to examine cardiovascular event risk associated with first-generation H1-antihistamine use in children and to provide valuable insights into their safety profile in real-world clinical practice. The findings of this study will contribute to informed decision-making regarding first-generation H1-antihistamine use in pediatric patients. 

## 2. Results

### 2.1. Baseline Characteristics

Of the eligible 5621 participants with cardiovascular events in the main cohort, 1992 discordant participants were prescribed first-generation antihistamines during the hazard or control period. Participant baseline characteristics on the index date are presented in Table 1. There were 348 (17.5%) children aged 6–24 months, 1194 (59.9%) aged 24 months–6 years, and 450 (22.6%) aged ≥7 years. There were 1010 (50.7%) male children. The numbers of children living in Seoul and in metropolitan, city, and rural areas were 499 (25.3%), 389 (19.8%), 942 (47.8%), and 139 (7.1%), respectively, which is similar to the residential area distributions for all children born between 2008 and 2009 [9]. The seasons at index date were 492 (24.7%) during spring, 511 (25.7%) in summer, 486 (24.4%) in fall, and 503 (25.3%) in winter. In terms of comorbidities, 119 (6.7%) children had obesity and 40 (2.0%) had Kawasaki disease. Acute bronchitis was the most frequently diagnosed (1087, 54.6%) condition in the hazard and control periods, followed by acute tonsillitis (284, 14.3%) and acute pharyngitis (283, 14.2%). Additionally, nasal decongestants and second-generation H1-antihistamines were used by 959 (48.1%) and 395 (19.8%) children, respectively, during the hazard and control periods.

In the second group, a total of 1387 patients were included as discordant participants. Among these participants, 214 children (15.4%) were aged 6–24 months, 861 children (62.1%) were aged 24 months–6 years, and 312 children (22.5%) were ≥7 years old. Overall, 704 participants (50.8%) were male children. The other baseline characteristics were comparable to those observed in the primary cohort. 

### 2.2. Association between Cardiovascular Events and First-Generation H1-Antihistamine

Table 2 shows first-generation H1-antihistamine use for each period before the index date and their association with cardiovascular events. In the main cohort, first-generation H1-antihistamine use was 1011 (50.8%) in the hazard period and 796 (40.0%) and 767 (38.5%) in the control periods. Cardiovascular event risk within 15 days of prescribing a first-generation H1-antihistamine was elevated (aOR [95% CI], 1.20 [1.13–1.27]). In sensitivity analyses conducted by setting the time window to 5 or 10 days, the elevated risks showed similar results (aOR [95% CI], 1.25 [1.17–1.33], or 1.250 [1.16–1.35], respectively).

In the second cohort, excluding children with specific birth histories and cardiovascular diseases, 694 (50.0%) children were exposed to first-generation H1-antihistamines in the hazard period, while 560 (40.4%) and 541 (39.0%) children were exposed during the control periods. The risk of cardiovascular events within 15 days of prescribing a first-generation H1-antihistamine drug was increased, and the results were comparable with those of the main cohort (aOR [95% CI], 1.19 [1.11–1.27]). Additionally, sensitivity analyses using a time window of 5 and 10 days in the second cohort also showed that first-generation H1-antihistamine use was associated with an increased risk of cardiovascular events (aOR [95% CI], 1.24 [1.13–1.35] and 1.26 [1.17–1.37], respectively).

Furthermore, after limiting the study to participants who were diagnosed with respiratory infections during the same period (hazard and control periods), prescribing first-generation H1-antihistamines within the hazard periods was still associated with an increased cardiovascular event risk (aOR [95% CI], 1.20 [1.13–1.27]) (all results are shown in Appendix A). 

Table 3 shows cardiovascular event risk according to participant characteristics. When the age at initial diagnosis of cardiovascular events was divided into 6–24 months, 2–6 years, and ≥7 years, all cardiovascular event risks owing to first-generation H1-antihistamine use were significant (aOR [95% CI], 1.28 [1.10–1.50], 1.20 [1.11–1.29], and 1.16 [1.03–1.31], respectively) and did not differ with age (*p* value for interaction > 0.05). Additionally, there were similar cardiovascular event risks between male (aOR [95% CI], 1.21 [1.11–1.31]) and female (aOR [95% CI], 1.19 [1.10–1.30]) children. Most of the cardiovascular events were arrhythmia (1902, 95.5%); however, the risks of arrhythmia and ischemic heart disease were both significant (aOR [95% CI], 1.19 [1.12–1.27] and 1.41 [1.06–1.88], respectively) and not different (*p* value for interaction = 0.25). Interestingly, depending on the season in which the cardiovascular events occurred, cardiovascular event risk increased during spring, fall, and winter (aOR [95% CI], 1.26 [1.11–1.41], 1.48 [1.31–1.67], and 1.15 [1.02–1.29], respectively), but not during summer (aOR [95% CI], 0.99 [0.88–1.11]).

## 3. Discussion

This general population-based, nationwide, administrative case-crossover study examined cardiovascular event risk associated with first-generation H1-antihistamine use in children. Our findings indicate that there is a 20% increased risk of cardiovascular events, especially arrhythmias, within 15 days of administering first-generation H1-antihistamines to children. When the time window for first-generation H1-antihistamine use was set to 10 or 5 days, cardiovascular event risk increased. Additionally, when children with specific birth histories and underlying cardiovascular diseases were excluded, first-generation H1-antihistamine drugs were associated with an increased risk of cardiovascular events. 

In the 1990s, cases of cardiovascular toxicities related to the administration of H1-antihistamine were reported and have since caught the attention of researchers. Several cases of cardiovascular toxicity induced by hydroxyzine have been reported; however, these events occurred when hydroxyzine was administered at substantial doses or to susceptible individuals [10,11,12]. Additionally, 3.81 patients with QT prolongation or TdP per million hydroxyzine usage years were found in the pharmacovigilance database, and only one case was reported in children aged <18 years [13]. Furthermore, cyclizine and dimethylene were reported to increase the risk of ventricular arrhythmia in a nested case-control study in Europe [14]. A recent pharmacovigilance study by the Food and Drug Administration Adverse Event Reporting System showed that other first-generation H1-antihistamines, including chlorpheniramine, clemastine, diphenhydramines, meclizine, promethazine, and trimethazine, are associated with an increased TdP risk [7].

Studies on the effects of first-generation antihistamines on cardiovascular event risk, including arrhythmias, are scarce. However, in two case-control and observational studies examining the cardiovascular side effects of second-generation antihistamines, an increased risk was reported [15,16]. On the other hand, a placebo and positive controlled, four-way crossover randomized trial showed that levocetirizine did not increase the risk of prolongation of the QT/QTc interval in healthy participants [17]. Additionally, a recent animal study on sensitive dogs and monkeys showed that levocetirizine had no effect on the QTc interval at high exposure levels that exceeded the supratherapeutic dose [18].

H1-antihistamines, despite their primary action on the H1-receptor, can induce cardiovascular toxicity that is not directly mediated by H1-receptor blockade. This suggests that the cardiovascular effects of H1-antihistamines are not class effects, implying that they can vary among different drugs within a class. Although the exact mechanisms of cardiovascular toxicity induced by H1-antihistamines are not fully understood, several explanations have been proposed. One of the proposed mechanisms suggests that H1-antihistamines may interfere with the function of cardiac potassium channels, specifically the I_Kr_ (rapidly activating component of delayed rectifier potassium current) component of cardiac repolarization currents [19,20]. An in vitro study showed that diphenhydramine blocks the repolarization of I_Kr_ currents, and other H1-antihistamines also have this effect [21,22]. This interference can induce a delay in cardiac repolarization, which may cause syncope, seizures, ventricular arrhythmias, and sudden death. 

The second potential explanation for the cardiovascular toxicity of H1-antihistamines is their anti-muscarinic properties. Certain first-generation antihistamines, such as diphenhydramine and chlorpheniramine, exert anticholinergic effects by blocking muscarinic acetylcholine receptors. Antimuscarinic activity in the heart causes pacemaker rhythm impairment, thereby inducing tachyarrhythmia [23,24]. Muscarinic receptors are found in various cardiac tissues, including the sinus node (SA) and the atrioventricular node (AV), and play important roles in heart rate and conduction control. By blocking these receptors, the antimuscarinic properties of H1-antihistamines can interfere with the normal function of the SA and AV nodes, thereby interfering with the normal electrical impulses that control the heartbeat [25]. These interruptions can cause tachycardia and tachyarrhythmias.

Additionally, an in vitro study suggested another potential mechanism linking H1-antihistamines and cardiac arrhythmias. The study found that H1-antihistamines reduced the expression levels of *pak1* (p21-activated kinase 1) mRNA and protein. *pak1* is an enzyme that plays a role in regulating ion channel activities, including those of cardiac ion channels [26]. By reducing *pak1* expression levels, H1-antihistamines may interfere with the normal function of ion channels involved in cardiac electrical activity. These changes can disrupt the process of repolarization, which affects the duration and pattern of action potentials in cardiac cells. Altered ion channel activity may also contribute to abnormalities in cardiac repolarization and increase the risk of arrhythmia.

Although these proposed mechanisms provide a possible explanation for the link between H1-antihistamines and cardiac arrhythmia, the overall clinical significance of real-world scenarios and certain risks remain topics of ongoing research. Healthcare providers should continue to consider and pay attention to the available evidence, especially for individuals with existing heart disease or other risk factors for arrhythmia, when prescribing H1-antihistamines.

To the best of our knowledge, this is the first nationwide general pediatric population study to evaluate the risk of cardiovascular events associated with the use of first-generation H1-antihistamines. Cases of cardiovascular adverse events associated with the administration of H1-antihistamines are rare, making it challenging to study them in prospective observational cohort studies or randomized trials, especially in children. This study serves as an ideal solution, addressing both the need for investigation and practical challenges simultaneously. 

This study has several limitations. These limitations are important for understanding the potential implications and generalizability of the study findings. The definition of cardiovascular events relied only on the ICD-10 codes for administrative data. Because these administrative data were established for national health insurance claims and not for research purposes, discrepancies may exist between the actual diagnosis and claims data. However, the definitions of cardiovascular events based on the ICD codes have been validated in previous studies [27,28]. Additionally, cardiovascular events were defined as ischemic heart disease and arrhythmia. Although approximately 95% of the cardiovascular events retrieved were arrhythmias, information on the detailed diagnoses and characteristics of these arrhythmias cases was limited. Secondly, we defined the use of first-generation H1-antihistamines based on prescription data. Therefore, it is unknown whether the drugs were actually administered to the patients. Third, it has been reported that respiratory viral infections, such as respiratory syncytial virus, influenza virus, and COVID-2019, are associated with an increased risk of arrhythmia [29,30,31,32,33]. Therefore, we attempted to confirm the associations after limiting the study participants to those with respiratory infections during the study period, and the associations were comparable with the main results (Appendix A). 

## 4. Methods

### 4.1. Data Source and Ethical Consideration

We obtained data from the National Health Insurance Service (NHIS) [9]. The NHIS, a single insurance system, achieved nearly 98% coverage for the entire population by 1989. The NHIS database contains information on demographic characteristics (age, sex, residential area, and health insurance premium) and healthcare utilization, including diagnoses (International Classification of Diseases, 10th version [ICD-10 codes]), prescribed medications, and procedures. Additionally, the National Health Checkup for Infants and Toddlers (NHSPIC) is a comprehensive health checkup program consisting of seven rounds of checkups for all citizens aged 4–72 months. It includes a general health questionnaire, developmental screening, oral health questionnaire, oral examination, anthropometric examination, physical examination, and age-specific anticipatory guidance. Detailed information regarding the NHIS database has been described previously [9]. The cohort was followed until 31 December 2018, or until participants were disqualified from health care due to mortality. The use of de-identified individual data for research purposes was authorized under the current National Health Insurance Act. The study protocol was reviewed and approved by the institutional review board of the Korea National Institute for Bioethics Policy (P01-201603-21-005).

### 4.2. Participants

A flowchart of the participant selection process is shown in Figure 1. Of the 2,395,966 children born in Korea between 1 January 2008 and 31 December 2012, we identified 6150 children who were diagnosed with cardiovascular events after 6 months of age. Out of these children, 5621 who were prescribed first-generation antihistamines prior to the date of cardiovascular event diagnosis were included in the main cohort.

A second cohort was established to minimize bias in the baseline health status of the participants, which could increase cardiovascular event risk. After excluding children with specific birth histories (birth weight <2.5 kg or >4 kg, premature birth, and multiple births), chromosomal anomalies, and underlying cardiovascular diseases (congenital heart disease, rheumatic heart disease, hypertensive disease, pulmonary heart disease, pulmonary circulation disease, pericarditis, endocarditis, myocarditis, non-rheumatic valvular disease, cardiomyopathy, and other heart diseases), 3968 children were finally included.

### 4.3. Case Cross-Over Study Design

We used a case-crossover study approach to evaluate the association between first-generation antihistamines and acute cardiovascular events. This case-crossover design was suitable for evaluating the effects of transient exposure to an acute event [34,35]. 

In this study, the date of initial diagnosis of the main cardiovascular event was defined as the index date. First-generation antihistamine exposure within 15 days preceding the index date (hazard period) was compared with exposure to first-generation antihistamines from 45–60 days and 75–90 days prior to the index date (the control periods, which indicate periods irrelevant to the cardiovascular event) in the same individual (Figure 2). Each participant acting as their own control in this within-participant comparison design exempted any influence of measured or unmeasured time-invariant confounders, such as family history or lifestyle factors, among participants [36]. The study findings were reported in keeping with the recommended guidelines for observational studies that use routinely collected health data (Appendix A). 

We compared exposure to first-generation antihistamines during the 0–15 days (time window = 15 days) before the cardiovascular event (hazard period) with exposure during the 45–60 and 75–90 days before the cardiovascular event (control period). Additionally, a time window of 10 or 5 days was used for sensitivity analysis. 

### 4.4. Cardiovascular Events

We assessed cardiovascular event risk from first-generation H1-antihistamine use (a summary of previous studies and literature searches is shown in Appendix A, respectively). The cardiovascular events of interest in our study were arrhythmias and ischemic heart disease. Cardiovascular events were ascertained using the ICD-10 codes of the claims data. Arrhythmia was defined as the first main diagnosis, with ICD-10 codes I44.X (atrioventricular and left bundle branch block), I45.X (conduction disorders), I47.X (paroxysmal tachycardia), I48.X (atrial fibrillation and flutter), and I49.X (other cardiac arrhythmias). Ischemic heart disease was defined as the first diagnosis based on ICD-10 codes I20.X (angina pectoris), I21.X (acute myocardial infarction), or I22.X (subsequent myocardial infarction). 

### 4.5. First-Generation H1-Antihistamine Use

Prescriptions of first-generation H1-antihistamine drugs (chlorpheniramine maleate, piprinhydrinate, hydroxyzine hydrochloride, phenylephrine hydrochloride, oxatomide, and mequitazine) were ascertained using the claims data. These drugs are the first-generation H1-antihistamines approved by the Korean Food and Drug Administration. If the prescription date of the first-generation antihistamine was included in the hazard or control period, it was defined as first-generation antihistamine use. 

### 4.6. Covariates

Information on the covariates in the database and the administrative codes are provided in Appendix A. Residential areas were divided into four groups: Seoul, metropolitan (Busan, Daegu, Incheon, Gwangju, Daejeon, and Ulsan), urban, and rural. Economic status was dichotomized according to insurance payment amounts. Data on birth weight and obesity were acquired from the NHSPIC [9]. Birth weight was recorded using the parent-answered NHSPIC questionnaire at 4–6 months of age. If diagnosed before the index date, Kawasaki disease, sleep apnea, hyperthyroidism, and diabetes mellitus were included as comorbidities. Obesity was defined as a body mass index z-score ≥1.64, which was calculated using the weight and height measured at the time closest to the index date (dividing body weight [kg] by the square of height [m]^2^). To account for within-participant time-variant covariates, concomitant diseases and medications were defined as diagnoses and prescriptions in the hazard and control periods for each participant. These concomitant conditions included acute nasopharyngitis, acute tonsillitis, acute pharyngitis, acute upper respiratory infection, acute sinusitis, acute suppurative otitis media, acute bronchitis, acute bronchiolitis, gastroenteritis, colitis, and allergic rhinitis. The concomitant medications included second-generation antihistamines, nasal decongestants, and systemic steroids.

### 4.7. Statistical Analysis

During the analysis of this case-crossover study, discordant participants who were exposed to first-generation H1-antihistamines in either the hazard or control periods, but not in both periods, contributed to the odds ratio (OR) estimation. ORs were used to measure the association between first-generation antihistamine exposure and cardiovascular events by comparing the exposure status of the participants during the hazard period with their own exposure status during the control periods. We used multivariable conditional logistic regression to estimate ORs and their 95% confidence intervals (CIs) using the LOGISTIC procedure in SAS software version 9.4 (SAS Institute Inc., Cary, NC, USA). All analyses were adjusted for age, sex, residential area, economic status, season at index date, comorbidities, concomitant diseases, and medications. The OR was interpreted as the incidence rate ratio.

For the sensitivity analysis, we considered pre-specified time windows (10 or 5 days), which means that the hazard period was 0–10 or 0–5 days before the cardiovascular event, and the corresponding control periods were 45–55, 45–50, 75–85, or 75–80 days before the cardiovascular events (Figure 2). Furthermore, we performed subgroup analyses stratified according to individual characteristics, including age, sex, residential area, economic status, season at index date, cardiovascular event type (arrhythmia or ischemic heart disease), specific birth history, and underlying cardiovascular disease. All analyses were adjusted for the same covariates as in the main analysis, except for the variables used for stratification. For intra-subgroup OR comparisons, the log-OR difference between the strata was used to calculate z-scores and *p* values.

Moreover, because respiratory infection itself increases cardiovascular disease risk, we further selected children who were diagnosed with respiratory infection during the hazard and control periods, even though concomitant respiratory infection risk was corrected when analyzing cardiovascular events in the main analysis. 

To minimize the type I errors of multiple comparisons, we predefined two-tailed *p*-values of <0.01 as statistically significant.

## 5. Conclusions

This nationwide and general pediatric population case-crossover study showed that first-generation H1-antihistamine use is associated with an increased cardiovascular event risk, especially arrhythmias, in young children. Cardiovascular toxicity is a significant concern in cases of H1-antihistamine overdoses. Therefore, our study highlights the importance of clinicians exercising utmost care when prescribing first-generation H1-antihistamines to children. 

## Figures and Tables

**Figure 1 pharmaceuticals-16-01073-f001:**
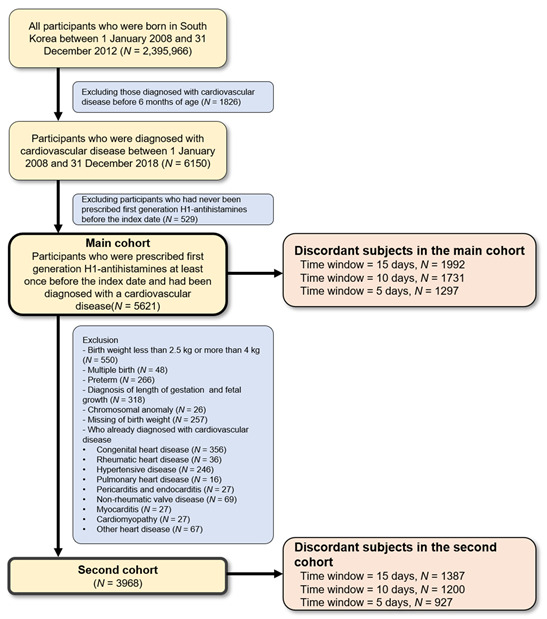
Participants selection process in a case-crossover study of first-generation antihistamine drug use and cardiovascular events.

**Figure 2 pharmaceuticals-16-01073-f002:**
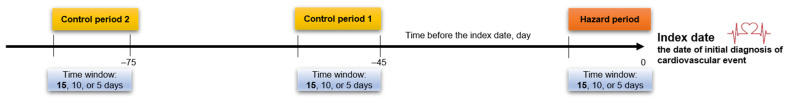
Case-crossover study design.

**Table 1 pharmaceuticals-16-01073-t001:** Baseline characteristics of the study population at the index date.

Variables ^a^	*N* (%)	
	Main Cohort ^b^ (*N* = 1992)	Second Cohort ^b^ (*N* = 1387)
Age		
6–24 months	348 (17.5)	214 (15.4)
24 months–6 years	1194 (59.9)	861 (62.1)
≥7 years	450 (22.6)	312 (22.5)
Sex		
Male	1010 (50.7)	704 (50.8)
Female	982 (49.3)	683 (49.2)
Residential area ^c^		
Seoul	499 (25.3)	360 (26.2)
Metropolitan	389 (19.8)	269 (19.6)
City	942 (47.8)	651 (47.4)
Rural	139 (7.1)	92 (6.7)
Economic status ^d^		
Below median	876 (46.0)	601 (45.4)
Median or over	1027 (54.0)	723 (54.6)
Season ^e^		
Spring (March–May)	492 (24.7)	331 (23.9)
Summer (June–August)	511 (25.7)	342 (24.7)
Fall (September–November)	486 (24.4)	359 (25.9)
Winter (December–February)	503 (25.3)	355 (25.6)
Comorbidities		
Kawasaki disease	40 (2.0)	28 (2.0)
Obesity	119 (6.7)	82 (5.9)
Sleep apnea	2 (0.1)	2 (0.1)
Hyperthyroidism	3 (0.2)	3 (0.2)
Diabetes mellitus	1 (0.1)	1 (0.1)
Concomitant diseases ^f^		
Acute nasopharyngitis	270 (13.6)	181 (13.0)
Acute tonsillitis	284 (14.3)	202 (14.6)
Acute pharyngitis	283 (14.2)	199 (14.3)
Acute upper respiratory infection	243 (12.2)	165 (11.9)
Acute sinusitis	193 (9.7)	136 (9.8)
Acute suppurative otitis media	122 (6.1)	83 (6.0)
Acute bronchitis	1087 (54.6)	765 (55.2)
Acute bronchiolitis	160 (8.0)	109 (7.9)
Gastroenteritis and colitis	108 (5.4)	85 (6.1)
Allergic rhinitis	229 (11.5)	166 (12.0)
Concomitant medication ^f^		
Second-generation antihistamine	395 (19.8)	301 (21.7)
Nasal decongestant	959 (48.1)	691 (49.8)
Systemic steroid	402 (20.2)	282 (20.3)

^a^ Variables were assessed on the date the participants were diagnosed with cardiovascular events—the index date. ^b^ Discordant participants when the time window was set to 15 days in the main and second cohorts. ^c^ Residential areas were divided into four groups: Seoul, metropolitan (Busan, Daegu, Incheon, Gwangju, Daejeon, and Ulsan), city, and rural. ^d^ Economic status was dichotomized according to the amount of insurance co-payment. ^e^ Season indicates the season at index date. ^f^ Concomitant disease and medication were defined as diagnosis and prescription in the hazard and control periods for each participant to be within-participant time-variant covariates.

**Table 2 pharmaceuticals-16-01073-t002:** Cardiovascular event risk associated with first-generation antihistamine use.

Time Window before Index Date	Total *N*	*N* (%)			Crude OR (95% CI)	Adjusted OR (95% CI) ^a^
		Exposed during the Hazard Period	Exposed during the Control Period 1	Exposed during the Control Period 2		
Main cohort						
0–15 days	1992	1011 (50.8)	796 (40.0)	767 (38.5)	1.203 (1.139–1.270)	1.201 (1.132–1.273)
0–10 days	1731	870 (50.3)	647 (37.4)	626 (36.2)	1.237 (1.167–1.312)	1.249 (1.170–1.333)
0–5 days	1297	639 (49.3)	452 (34.8)	445 (34.3)	1.254 (1.172–1.341)	1.250 (1.157–1.351)
Second cohort						
0–15 days	1387	694 (50.0)	560 (40.4)	541 (39.0)	1.181 (1.107–1.260)	1.189 (1.110–1.274)
0–10 days	1200	607 (50.5)	450 (37.5)	431 (35.9)	1.248 (1.164–1.339)	1.264 (1.171–1.365)
0–5 days	927	457 (49.3)	319 (34.4)	326 (35.2)	1.238 (1.143–1.341)	1.235 (1.129–1.352)

Abbreviations: *N*, number; OR, odds ratio; CI, confidence interval. ^a^ Adjusted for sex, age, residential area, economic status, season, comorbidities, concomitant diseases, and concomitant medication use.

**Table 3 pharmaceuticals-16-01073-t003:** Effects of antihistamine drug use on cardiovascular event risk stratified by individual characteristics in the main cohort.

Variables ^a^	Total *N* = 1992		
	*N* (%)	Adjusted OR (95% CI) ^b^	*p* Value
Age			
6–24 months	348 (17.5)	1.283 (1.095–1.503)	Ref
24 months–6 years	1194 (59.9)	1.199 (1.113–1.292)	0.390
≥7 years	450 (22.6)	1.159 (1.028–1.305)	0.244
Sex			
Male	1010 (50.7)	1.207 (1.111–1.312)	Ref
Female	982 (49.3)	1.194 (1.098–1.298)	0.829
Residential area ^c^			
Seoul	499 (25.3)	1.229 (1.095–1.380)	Ref
Metropolitan	389 (19.8)	1.111 (0.973–1.269)	0.165
City	942 (47.8)	1.172 (1.077–1.275)	0.439
Rural	139 (7.1)	1.746 (1.346–2.265)	0.039
Economic status ^d^			
<Median	876 (46.0)	1.273 (1.166–1.390)	Ref
≥Median	1027 (54.0)	1.144 (1.057–1.239)	0.04
Season			
Spring (March–May)	492 (24.7)	1.255 (1.114–1.413)	Ref
Summer (June–August)	511 (25.7)	0.990 (0.883–1.110)	<0.001
Fall (September–November)	486 (24.4)	1.482 (1.313–1.674)	0.045
Winter (December–February)	503 (25.3)	1.145 (1.017–1.290)	0.200
Cardiovascular events ^e^			
Arrhythmia	1902 (95.5)	1.192 (1.122–1.265)	Ref
Ischemic heart disease	90 (4.5)	1.414 (1.063–1.882)	0.245
With specific birth history ^f^			
No	1639 (82.3)	1.193 (1.120–1.271)	Ref
Yes	353 (17.7)	1.252 (1.066–1.470)	0.526
With underlying cardiovascular disease ^g^			
No	1658 (83.2)	1.208 (1.132–1.288)	Ref
Yes	334 (16.8)	1.165 (1.007–1.347)	0.584

Abbreviations: *N*, number; OR, odds ratio; CI, confidence interval; Ref, reference. ^a^ Variables were assessed on the date participants were diagnosed with a cardiovascular event—the index date. ^b^ Adjusted for sex, age, residential area, economic status, season, comorbidities, concomitant disease, and concomitant medication use. ^c^ Residential areas were divided into four groups: Seoul, metropolitan (Busan, Daegu, Incheon, Gwangju, Daejeon, and Ulsan), city, and rural. ^d^ Economic status was dichotomized according to the amount of insurance co-payment. ^e^ Cardiovascular events were defined as the main diagnosis of arrhythmia or ischemic heart disease. Arrhythmia had ICD-10 codes I44.X (atrioventricular and left bundle branch block), I45.X (conduction disorders), I47.X (paroxysmal tachycardia), I48.X (atrial fibrillation and flutter), or I49.X (other cardiac arrhythmias). Ischemic heart disease was classified as ICD-10 codes I20.X (angina pectoris), I21.X (acute myocardial infarction), or I22.X (subsequent myocardial infarction). ^f^ Specific birth history indicated birth weight <2.5 kg or >4 kg, multiple births, prematurity, diagnosis of ICD-10 codes P05.X–P08.X (length of gestation and fetal growth), and a chromosomal anomaly. ^g^ Underlying cardiovascular disease indicates ICD-10 codes of Q20.X–Q28.X (congenital heart disease), I00.X–I09.X (rheumatic heart disease), I10.X–I13.X (hypertensive disease), I26.X–I28.X (pulmonary heart disease and pulmonary circulation disease), I30.X–I32.X (pericarditis), I34.X–I37.X (non-rheumatic valve disease), I33.X, I38.X–I41.X (endocarditis and myocarditis), I42.X–I43.X (cardiomyopathy), and I51.X–I52.X (other heart diseases).

## Data Availability

This study was based on the National Health Claims Database (NHIS-2019-1-560) established by the NHIS of the Republic of Korea. Applications using NHIS data are reviewed by the Inquiry Committee of Research Support, and if the application is approved, raw data are provided to the applicant for a fee. We cannot provide access to the data, analytical methods, and research materials to other researchers because of the intellectual property rights of this database, which is owned by the National Health Insurance Corporation. However, investigators who wish to reproduce our results or replicate the procedure can use the database, which is available for research purposes (https://nhiss.nhis.or.kr/ accessed on 30 June 2023).

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
