# Peer review of "Association between First-Generation Antihistamine Use in Children and Cardiac Arrhythmia and Ischemic Heart Disease: A Case-Crossover Study"

_pharmaceuticals, 2023, doi:10.3390/ph16081073_

Round 1

Reviewer 1 Report

The manuscript describes the results of a case-crossover study that examines the association between first-generation H1-antihistamines and cardiovascular events in children born between 2008 and 2012 in Korea. The study compares the use of these medications during the hazard period (0-15 days before the cardiovascular events) with usage during the control periods (45-60 and 75-90 days before the events). The study found that the risk of cardiovascular events was increased among children using these medications, and the association remained significant even within shorter time windows. Although the paper is nicely organized, there were still some issues to be improved.

1. In the Introduction, the core contents of the manuscript haven’t been presented clearly. For instance, 1) what is the existed question? 2) what is your solution? 3) what is the expected outcome, and what is potential impact of this study? The content of the introduction is too simple.

2. The conclusion is not clear. The authors combined a large amount of clinical data and statistical methods, and concluded that first-generation antihistamines may increase the risk of cardiovascular disease. Any medication can have side effects, but the view that it increases the risk of cardiovascular disease is too broad, and lacks specificity and clarity.

3. Please provide detailed information in the methods section on which statistical methods were used in the study and their specific steps.

Reviewer 2 Report

The authors presented a study on the risk of cardiovascular events among children using first-generation H1-antihistamines. In my opinion, the study prepared is interesting but does not show the current state of knowledge for 2023. My comments and suggestions:

1. The "References" section of the article is deficient, containing only 29 literature items, the most recent of which are from 2021.

2. The authors often write "a recent study" and cite a study from, for eg, 2016 (page 11). Such information misleads the reader.

3. The article gives the impression that it was prepared in 2021, so the 'a case-crossover study' should be updated with data published up to 2023.

4. The article was prepared carelessly with disregard for Instructions for Authors.

Author Response

Point-by-point response to reviewer’ 2 comment

General comment: The authors presented a study on the risk of cardiovascular events among children using first-generation H1-antihistamines. In my opinion, the study prepared is interesting but does not show the current state of knowledge for 2023. My comments and suggestions:

Comment:

  1. The "References" section of the article is deficient, containing only 29 literature items, the most recent of which are from 2021.
  2. The authors often write "a recent study" and cite a study from, for eg, 2016 (page 11). Such information misleads the reader.
  3. The article gives the impression that it was prepared in 2021, so the 'a case-crossover study' should be updated with data published up to 2023.
  4. The article was prepared carelessly with disregard for Instructions for Authors.

Reply:  Thank you for your valuable time and effort in reviewing our manuscript. We apologize for the lack of recent references in this paper. In line with your feedback, we have made the necessary revisions to the manuscript. We have reviewed the literature and included relevant updates. Specifically, we have replaced older references with more recent ones and also included several additional recent references. Furthermore, we have thoroughly discussed relevant papers that were published in 2023 within the discussion section. These updates ensure that the paper incorporates the latest research findings and provides a comprehensive analysis of the subject matter. Furthermore, we have made the necessary adjustments to align the paper with the formatting guidelines of the journal.

Revised references

  1. Shahn, Z.; Hernán, M.A.; Robins, J.M. A formal causal interpretation of the case‐crossover design. Biometrics 2023, 79, 1330-1343.
  2. Engwall, M.J.; Baublits, J.; Chandra, F.A.; Jones, Z.W.; Wahlstrom, J.; Chui, R.W.; Vargas, H.M. Evaluation of levocetirizine in beagle dog and cynomolgus monkey telemetry assays: Defining the no QTc effect profile by timepoint and concentration‐QTc analysis. Clinical and Translational Science 2023, 16, 436-446.
  3. Kawashima, H.; Inagaki, N.; Nakayama, T.; Morichi, S.; Nishimata, S.; Yamanaka, G.; Kashiwagi, Y. Cardiac Complications Caused by Respiratory Syncytial Virus Infection: Questionnaire Survey and a Literature Review. Global Pediatric Health 2021, 8, 2333794X211044114.
  4. Song, M.K.; Kwon, B. Arrhythmia and COVID-19 in children. Clin Exp Pediatr 2023, 66, 190-200, doi:10.3345/cep.2023.00024.
  5. Lee, P.Y.; Garan, H.; Wan, E.Y.; Scully, B.E.; Biviano, A.; Yarmohammadi, H. Cardiac arrhythmias in viral infec-tions. Journal of Interventional Cardiac Electrophysiology 2023, 1-15.

[original manuscript, page 11]

Additionally, a recent in vitro study has suggested another potential mechanism linking H1-antihistamines and cardiac arrhythmias.

  • [Revised manuscript, page 11, line 356]

Additionally, an in vitro study suggested another potential mechanism linking H1-antihistamines and cardiac arrhythmias.

Round 2

Reviewer 1 Report

 The authors have adeqautely addressed all my remarks and the improved paper can be now accepted in the present form.

Reviewer 2 Report

The article has been mostly corrected by the authors and can be published.